# Obesity-Related Brain Cholinergic System Impairment in High-Fat-Diet-Fed Rats

**DOI:** 10.3390/nu14061243

**Published:** 2022-03-15

**Authors:** Ilenia Martinelli, Seyed Khosrow Tayebati, Proshanta Roy, Maria Vittoria Micioni Di Bonaventura, Michele Moruzzi, Carlo Cifani, Francesco Amenta, Daniele Tomassoni

**Affiliations:** 1School of Pharmacy, University of Camerino, 62032 Camerino, Italy; ilenia.martinelli@unicam.it (I.M.); khosrow.tayebati@unicam.it (S.K.T.); mariavittoria.micioni@unicam.it (M.V.M.D.B.); michele.moruzzi@unicam.it (M.M.); carlo.cifani@unicam.it (C.C.); francesco.amenta@unicam.it (F.A.); 2School of Biosciences and Veterinary Medicine, University of Camerino, 62032 Camerino, Italy; proshanta.roy@unicam.it

**Keywords:** obesity, brain, cholinergic system, high-fat diet, muscarinic receptors

## Abstract

A link between obesity and cerebral health is receiving growing recognition. Here, we investigate in the frontal cortex and hippocampus the potential involvement of cholinergic markers in brain alterations previously reported in rats with obesity induced by diet (DIO) after long-term exposure (17 weeks) to a high-fat diet (HFD) in comparison with animals fed with a standard diet (CHOW). The obesity developed after 5 weeks of HFD. Bodyweight, systolic blood pressure, glycemia, and insulin levels were increased in DIO rats compared to the CHOW group. Measurements of malondialdehyde (MDA) provided lipid peroxidation in HFD-fed rats. Western blot and immunohistochemical techniques were performed. Our results showed a higher expression of choline acetyltransferase (ChAT) and vesicular acetylcholine transporter (VAChT) in obese rats but not the VAChT expression in the frontal cortex after 17 weeks of HFD. Furthermore, the acetylcholinesterase (AChE) enzyme was downregulated in HFD both in the frontal cortex and hippocampus. In the brain regions analyzed, it was reported a modulation of certain cholinergic receptors expressed pre- and post-synaptically (alpha7 nicotinic receptor and muscarinic receptor subtype 1). Collectively, these findings point out precise changes of cholinergic markers that can be targeted to prevent cerebral injuries related to obesity.

## 1. Introduction

The prevalence of obesity has reached a global proportion. The most important cause of obesity is excessive food intake, and relative behavioral changes have recently been reported [1]. Adipose tissue accumulation and dysfunction characterize obesity. This also leads to an imbalance in adipocytokines production, which in turn triggers an uncontrolled chronic inflammation [2,3]. In addition, insulin resistance due to obesity is linked to wide metabolic abnormalities [4], which are associated with the development of severe pathologies [5].

The obese condition and long-term high-fat diet (HFD) consumption, for instance, diets rich in saturated fat, have been associated with less cognitive capacity in both humans and animals [6,7,8,9]. Population-based studies reported that the obesity-driven vascular risk factors, such as hypertension, dyslipidemia, and diabetes, are all connected with cognitive deterioration and neurodegenerative dementias, such as Alzheimer’s disease (AD), usually characterized by amyloid-aggregates forming plaques, loss of synaptic transmission and degeneration of cholinergic neurons [10,11]. Overlapping neurodegenerative mechanisms have been recognized in obese conditions: neuro-inflammation, oxidative stress, and mitochondrial dysfunction [2,11]. Then, alterations at the blood-brain barrier (BBB) integrity and peptides transport in obesity were observed [12]. Additionally, in animals, HFD influences negatively the cognitive activity, compromising the BBB integrity [13]. HFD-induced obesity is associated with insulin resistance and other chronic diet-related illnesses, including dementia [14]. Indeed, dietary composition as an environmental factor can negatively affect hippocampal activity. Many data have demonstrated in middle-aged rats that HFD negatively impacts the memory processes controlled by the hippocampus, involving learning and use of stimuli in the spatial environment [15,16].

In accordance recently, we reported in diet-induced obese (DIO) rats the presence of increased glycemia, insulin, and systolic blood pressure. In the hippocampus and frontal cortex, astrogliosis, activation of microglia, transient receptor potential (TRP) ion channels dysfunction, and endothelial inflammation have been found [17,18]. In addition, in the obese Zucker rats (OZRs), we reported BBB changes, loss of neurons, gliosis, as well as oxidative stress in the same brain regions [19,20]. Both microglia and astrocytes are critical for keeping BBB integrity, retaining neuronal metabolism in response to local tissue damage. Their activation in the hypothalamus of rats and humans follows HFD consumption [21]. These data were consistent with our recent findings, in which 20-week-old OZRs demonstrated cholinergic and synaptic alterations, highlighting that obesity and aging induced cognitive dysfunction [22]. A protective effect may be raised from the activation of the anti-inflammatory pathway involving the astroglial α7nicotinic acetylcholine receptors (α7nAChR) [23,24]. Cholinergic pathways project widely to different brain areas: basal-forebrain, cerebral cortex and hippocampus, as well as some intrinsic cholinergic hippocampal interneurons, show a decisive part in cognitive activity [25]. Among the consequences of AD, the presynaptic cholinergic hypofunction is the most relevant; thus, cholinergic replacement therapy is beneficial in alleviating the cognitive dysfunction. Even if acetylcholinesterase (AChE) inhibitors (donepezil, galantamine, or rivastigmine) stop the breakdown of synaptic acetylcholine (ACh), enhancing the cholinergic and cognitive functions, which deteriorate in normal aging and AD, they have some important limitations [26]. Furthermore, new insights into acetylcholine receptors (AChRs) molecular pharmacology are bringing other drugs directed at AChRs to the center stage [27].

Histochemical techniques show ChAT-immunoreactive fibers in layers III and IV of the cerebral cortex, with faint labeling in layer II. Many neurons positive to ChAT shown a bipolar dendritic pattern [28,29]. The hippocampus presents a finely granular distribution of neurons positive to ChAT in the pyramidal layer, with an intense immunoreactivity, especially, in the CA2 and CA3 subfields [29]. The rat hippocampus receives afferent fibers from the AChE, and they take part in cholinergic system activation [30]. A double labeling of both markers ChAT and AChE is not intense enough to be observed [29,31].

The neurobiological mechanisms, which cause brain impairment following HFD consumption, are not understood. However, it is still unknown whether HFD uniquely impairs the frontal cortex and the hippocampus, brain regions that are important for cognition. Despite the well-known global impact of overweight and obesity in the incidence of brain dysfunction, many aspects of this connection, such as the involvement of the cholinergic pathway, are still inconsistently defined [32]. Based on this evidence, in our study, we aimed at founding such a connection by assessing the effects of obesity on brain cholinergic signaling, using a rodent model of diet-induced obesity (DIO), which mimics mutual features of human obesity more precisely than other genetic models [33,34]. The study was performed in cerebral zones, where the cholinergic system is implicated: the frontal cortex, particularly the motor region, and the hippocampus, which is concerned in memory and learning skills [20]. To investigate possible mechanisms underlying these disturbances, we examined the expression of cholinergic markers also involved in neurodegenerative diseases [35], such as choline acetyltransferase (ChAT) and AChE, the enzymes used to synthesize and break down the ACh, respectively. In addition, the levels of the vesicular acetylcholine transporter (VAChT), which mediates the packaging and transport of ACh, were examined. Different AChRs subtypes 1, 3, and 5 muscarinic acetylcholine receptors (mAChR1, mAChR3, and mAChR5, respectively) were explored because of their ubiquity and richness in the cerebral cortex and the hippocampus [36,37], and they are all mainly postsynaptic and Gq coupled [38,39]. Moreover, the nicotinic alpha-7 receptor (α7nAChR), predominant and widely expressed at pre-and postsynaptic levels [40], was analyzed.

## 2. Materials and Methods

### 2.1. Animals

Wistar rats (Charles River; male; *n* = 36; body weight = 225–250 g) were used. All procedures concerning animals followed the Institutional Guidelines and conformed with the Italian Ministry of Health (protocol number 1610/2013) and associated guidelines from the European Communities Council Directive. Animals were arbitrarily separated into two groups: CHOW (*n* = 16) with standard diet *ad libitum* (4RF18, Mucedola, Settimo Milanese, Italy; 2.6 kcal/g, which was composed among carbohydrates, proteins, and fats as follows: 72%, 21%, and 7%, respectively) and DIO (*n* = 20) with HFD *ad libitum* (D12451, Research Diets, Inc., New Brunswick, NJ; 4.73 kcal/g, distributed in 45% fat, 35% carbohydrate, 20% protein;). After 5 weeks (12 weeks of age) of HFD, the obese phenotype was established, and DIO (*n* = 8) and CHOW (*n* = 8) rats were sacrificed. The remaining DIO rodents were fed with HFD for further 12 weeks (24 weeks of age), for a total of 17 weeks of HFD, while the remaining CHOW rats were fed with the standard diet [17,18,41,42,43]. In the groups of DIO animals, after 17 weeks of HFD, rats that were resistant to increase of body weight (*n* = 3) were excluded from the experiment because they did not become obese [42,44,45,46,47]. Daily, food consumption and body weight were measured. Weekly systolic blood pressure was recorded.

### 2.2. Blood Parameters and Brain Tissue Preaparation

At sacrifice, systolic blood pressure was recorded. In blood withdrawals, the following parameters were estimated: glucose, insulin, triglycerides, and total cholesterol. The thiobarbituric-acid-reactive substances (TBARS) quantity, measured as the malondialdehyde (MDA), was evaluated in the serum. The assay kits and all these procedures were reported previously [17,18,41,42,43]. Finally, the brains were removed and divided in two hemispheres. The right was immediately frozen for biochemical analysis, while the left was fixated in 4% paraformaldehyde solution in PBS, in order to be embedded subsequently in paraffin wax.

### 2.3. Histochemistry, Immunohistochemistry and Image Analysis

With a microtome, the paraffin-embedded brains were cut into sagittal sections of 10 μm and mounted on microscope slides in order to perform the morphological and immunohistochemical (IHC) analysis, as previously described [17,18]. Consecutive sagittal sections were stained with a 0.75% cresyl violet solution (Nissl’s staining) to investigate brain gross anatomy and the presence of tissue degeneration. For IHC, primary antibodies were diluted (Table 1), and sections were incubated overnight at 4 °C. Through preliminary experiments, the specificity and the concentration of the antibodies were established [22,48,49,50,51]. The sections were incubated with the specific biotinylated secondary antibodies (Bethyl Laboratories, Inc., Montgomery, TX, USA, dilution 1:200). Sections were then washed in PBS and incubated with avidin-biotin-peroxidase complex (Vectastain ABC Elite kit; Vector Laboratories, Inc., Burlingame, CA, USA). Finally, the immunoreaction was revealed using a 3,3′-diaminobenzidine tetrahydrochloride (DAB) substrate kit (Vector Laboratories, Inc., Burlingame, CA, USA) according to the manufacturer’s protocol. The intensities of immune reaction were measured with NIS Elements Nikon (Florence, Italy) image analyzer software, and representative pictures were captured by a microscope Leica DMR coupled by DS-Ri2 NIKON camera [17,18,22]. To assess the immunostaining background, some sections were not incubated with primary antibody but with a non-immune serum.

### 2.4. Western Blot and Quantification

Brain tissues approximately 0.1 ± 0.02 g, were lysed in appropriate buffer with protease inhibitors (Sigma-Aldrich, Milan, Italy). Following centrifugation, for the Western blot procedures, the protein amount in the supernatant was quantified. A total of 40 µg of proteins was loaded in sodium dodecyl sulfate (SDS) polyacrylamide gel and transferred onto nitrocellulose membranes, which were incubated with the appropriate primary antibodies at different dilutions (Table 1) as previously established [17,22]. Then, specific secondary antibodies conjugated with horseradish peroxidase (HRP) (Bethyl Laboratories, Inc., Montgomery, TX, USA, dilution 1:5000) were used. After, protein bands were visualized using chemiluminescent substrates (EuroClone, Milan, Italy) followed by densitometric analysis. To normalize, monoclonal anti-beta-actin (β-actin) or glyceraldehyde-3-phosphate dehydrogenase (GAPDH) antibodies, produced both in mice, were used as a loading control.

### 2.5. Statistical Analysis

Data were analyzed using GraphPad prism software. The significance of difference between means was assessed by ANOVA. All the results were expressed as the mean ± standard error of mean (S.E.M.), with *p*-value < 0.05 being considered as statistical significance.

## 3. Results

### 3.1. Body Weight, Food Consumption, and Blood Parameters

No significant changes were observed between the groups at the beginning of the study in terms of body weight. As expected in the HF rats, we found an increase in body weight in DIO rats in comparison to CHOW rats and the obese phenotype was established at 5 weeks of diet. At the same time, systolic blood pressure, glycemia, total cholesterol, and triglycerides levels did not change except for insulin values, which were significantly higher in DIO rats with a mean value of 0.79 µg/L with a S.E.M. of 0.12 compared to CHOW (0.31 ± 0.06 µg/L; *p* < 0.05 vs. DIO rats). After 17 weeks, the body weight in DIO rats was remarkably higher than in CHOW rats. As already published [17,18,41,42,43], the DIO animals showed a significant increase in the systolic blood pressure, glycemia, and insulin levels after 17 weeks of HFD, in comparison to CHOW rats. The obesity did not influence neither the triglycerides nor the total cholesterol. In addition, the TBARS assay demonstrated an increased serum concentrations of MDA in DIO rats (26.0 ± 1.5 µM) compared with the control CHOW group (18.8 ± 1.8 µM; *p* < 0.05 vs. DIO rats) after 5 weeks of HFD. Previously, we reported a remarkably increased MDA concentration also after 17 weeks of HFD [41].

### 3.2. Neuronal Nuclei and Neurofilament

As shown in Appendix A, no microanatomical changes were observed in the frontal cortex (Appendix A) and hippocampus (Appendix A) of DIO rats after 5 and 17 weeks of HFD in comparison with age-matched CHOW rats. Histochemical analysis confirmed the magnetic resonance imaging (MRI) results that did not show vascular and morphological alterations in the older DIO rats [52]. Moreover, to evaluate neuronal viability, sections were immunostained, using antibodies against NeuN and NF to mark neurons. After 5 weeks of HFD, neither Western blot nor immunohistochemistry revealed differences in NeuN (Appendix A) as well as NF (Appendix A) among the experimental groups both in the frontal cortex and in the hippocampus. Even if NeuN levels and positive neurons were not modulated after longer-lasting exposure to HFD (Appendix A), NF levels diminished in the two brain areas analyzed of obese animals in comparison to controls (Appendix A).

### 3.3. Choline Acetyltransferase and Acetylcholinesterase

Immunochemical analyses pointed out an increase in the frontal cortex (Figure 1A) and the hippocampus (Figure 1C), in the ChAT levels at 68 kDa in HFD-fed rats after 5 and 17 weeks in comparison to controls. The immunohistochemistry analysis was in line with the proteins expression. The average intensity values of ChAT were significantly increased after the exposure to HFD in the frontal cortex (Figure 1B) as well as in the hippocampus (Figure 1D). Representative pictures showed that the immunoreaction was localized in the afferent fibers in layers III and IV of the cerebral cortex (Figure 1B) and in the subfields of the hippocampus, CA2 and CA3 (Figure 1D).

Western blot analysis for the AChE carried out in the frontal cortex (Figure 2A) and in the hippocampus (Figure 2C) showed a band at 70 kDa in both these areas. A decrease of AChE level was evident in obese rats fed for 17 weeks with HFD in comparison to controls (Figure 2A,C). The AChE immunoreaction was mainly limited in the neurons of the V layer of the frontal cortex (Figure 2B). Pyramidal neurons were reactive both in the hippocampal CA1 and in CA2 (Figure 2D). The immunoreaction of AChE was found to be remarkably less intense in DIO animals fed with HFD for 17 weeks in the two brain areas analyzed (Figure 2B,D).

### 3.4. Vesicular Acetylcholine Transporter

Immunochemical results for the VAChT presented a band at 80 kDa in the frontal cortex (Figure 3A) as well as in the hippocampus (Figure 3C).

The VAChT level was remarkably higher in the frontal cortex of DIO rats after 5 weeks of HFD. On the contrary, it was found after 17 weeks with HFD (Figure 3A). The immunohistochemistry analysis confirmed these results in the frontal cortex, where immunoreactivity was found in cell bodies of the V layer (Figure 3B). While differences in proteins levels were reported in the hippocampus only after 17 weeks of hypercaloric diet (Figure 3C), VAChT immunoreaction was significantly increased in the CA1 subfield (Figure 3D) of DIO rats both after 5 and 17 weeks of HFD exposure.

### 3.5. Alpha7 Nicotinic Acetylcholine Receptor

Western blot results for the α7nAChR showed a 55 kDa band in the two brain areas considered (Figure 4A,C). In the frontal cortex, protein quantification reported a less expression of α7nAChR in obese animals fed for 17 weeks with HFD compared with lean rats (Figure 4A), while in the hippocampus, a remarkable reduction was reported both after 5 and 17 weeks of hypercaloric diet (Figure 4C). The immunostaining for the nicotinic receptor α7nAChR was present in the neurons of the V layer of the frontal cortex (Figure 4B), the CA1, and the CA3 subfields (Figure 4D). The intensities of immunoreaction for α7nAChR confirmed the immunochemical results in these two brain areas analyzed (Figure 4B,D).

### 3.6. Muscarinic Acetylcholine Receptors

Among the different muscarinic receptors considered, the mAChR1 was altered in the obese phenotype (Figure 5). Immunochemical data for the mAChR1 displayed its expression at 50 kDa, that was significantly increased in 5 weeks’ HFD-fed rats compared to that in controls in the frontal cortex (Figure 5A) but not in the hippocampus (Figure 5C). The immunohistochemistry analysis for mAChR1 (Figure 5B,D) confirmed the Western blot data. mAChR1 was highly expressed on the bodies of pyramidal neurons in the V layer of the frontal cortex (Figure 5B) as well as in the CA3 subfield of the hippocampus (Figure 5D).

Besides, the mAChR3 receptor was expressed with an 80 kDa band in in the two brain regions considered (Figure 6A,C). Neither the mAChR3 expression (Figure 6A,C) nor its immunoreaction (Figure 6B,D) were remarkably changed between the opposite groups. As shown by representative pictures, mAChR3 was expressed on pyramidal neurons of the V layer of the frontal cortex (Figure 6B) as well as on the neurons of the CA3 subfield of the hippocampus (Figure 6D).

Finally, the mAChR5 receptor was expressed at approximately 55 kDa in the frontal cortex (Figure 7A) and hippocampus (Figure 7C). The expression of the mAChR5 subtype of DIO rat after 5 weeks or 17 weeks of HFD did not differ significantly compared to CHOW in the frontal cortex (Figure 7A) as well as in the hippocampus (Figure 7C). Similar results were obtained from the IHC (Figure 7B,D). mAChR5 was expressed in the pyramidal neurons of both the V layer of the frontal cortex (Figure 7B) and in the hippocampal CA3 subfield (Figure 7D).

## 4. Discussion

Obesity is a condition associated with numerous physiological abnormalities caused by fat tissue accumulation. The link between obesity and cognitive dysfunction has been established. Genetically, obese rats, such as the leptin-receptor-deficient OZRs, reported cognitive dysfunction related to BBB impairments, loss of neurons, inflammation, and gliosis accompanied by behavioral tests alterations [19,20]. In accordance, data showed that HFD led to compromised memory and learning in rodents [53,54,55]. As previously reported, 24-week-old DIO rats showed a condition of anxiety-like behavior and a decrease of retention latency time in the emotional learning skill if compared with CHOW rats [17]. Since ACh is expected to play a role in mediating the fluctuations in emotional behaviors induced by HFD intake [56], and the cholinergic activity has been found altered in older OZRs [22], here, we also investigated the possible modulation of the cholinergic pathway after exposure to HFD.

In obese or fatty Zucker rats, the results regarding the levels of ChAT and AChE enzymes were complex and controversial depending especially on the brain areas analyzed [57,58]. Since 14-week-old OZRs developed an enhanced both in ChAT and in AChE activities, it was postulated an intensification of the ACh turnover rate by [59], whereas in Zucker fatty rats, the lower ACh content was found in all brain regions, highlighting a dysfunction. In addition, lower ChAT activity as well as higher AChE activity was found only in the medulla oblongata and striatum of fatty rats [57]. Conversely, the reduced enzymatic activity of the AChE, particularly in the frontal cortex, hypothalamus, and midbrain area of rats fed with HFD, reflected a probable augmentation in ACh concentration in these cerebral areas [56]. These data were in agreement with [59], which reported a suppressive outcome on AChE action in the cortex and hypothalamus after long-term (6 months) HFD. Short-term (7 days) fat consumption was effective in stimulating the cholinergic system in cortical, hypothalamic, and midbrain regions [56]. As in vitro, the activity of AChE was reduced by the augmentation of free fatty acid in the HFD-consuming rats [60,61], and probably these circulating energy molecules influenced the cholinergic function [49]. Here, our results showed an earlier higher expression of ChAT and a later lower expression of AChE in obese animals both in the frontal cortex as well as in the hippocampus; perhaps it is an attempt of recovery mechanism to restore the ACh content or of increase in the ACh turnover or to positively control the NF levels, diminishing the axonal damage previously reported in DIO rats [17]. The alterations of axonal fibers could explain the downregulation of VAChT expression in the frontal cortex of 24-week-old DIO rats, similar to the VAChT modulation present both in the frontal cortex and hippocampus of older OZRs [22]. Differently, an upregulation of VAChT, as observed in spontaneously hypertensive rats, represents an attempt to compensate for cholinergic impairment in the first stages of brain dysfunction induced by hypertension [50,51].

Different anti-inflammatory pathways control the HFD-induced inflammation, including the cholinergic pathway [62], through the activation of the α7nAChR [63]. A stimulatory reaction of dietary fat has been reported on α7nAChR in the lateral and ventromedial hypothalamus [56]. Unlike other studies also in the hypothalamus, liver, spleen, and adipose tissue, we sustained that HFD reduced the expression of α7nAChR [64,65]. In accordance, our data demonstrated a reduction of this receptor both in the frontal cortex and hippocampus in rats fed with HFD, as we described recently in the same regions of older OZRs [22]. Since the stimulation of α7nAChR both in microglia as well as in astrocytes can induce neuroprotection resulting from anti-inflammatory activities [66], we could speculate that the reactive microglia, astrogliosis, and vascular inflammation, characterized by an increase of endothelial inflammatory markers (cell adhesion molecules) in DIO rats after 17 weeks of HFD [17], may be related to the low expression of α7nAChR in obese phenotype.

Among the muscarinic acetylcholine receptors (mAChRs), type 1 in the hippocampus and cerebral cortex plays a central role in cognitive processing, memory, and learning, and it is compromised in AD [67]. Recently, we reported that mAChR1 was remarkably downregulated in the hippocampus of OZRs without any differences in mAChR3 and mAChR5 levels [22]. The mAChRs were distinctly influenced by obesity in DIO rats [39,68]. Indeed, downregulation of both mAChR1 and mAChR3 in the hippocampus of DIO rats has been described without any differences in mAChR5 [68]. Interestingly, the current study showed that mAChR1 was upregulated after 5 days of HFD only in the frontal cortex. mAChR1 presence was increased in diabetic rats with a reduction in affinity [69]. The observed alterations in the receptor number and affinity were due to the changes of receptor protein and synthesis [69]. Here, no remarkable differences were detected in mAChR3 levels as well as for mAChR5. Collectively, the present data suggest a differential variation of muscarinic subtypes in obese rats compared to lean controls.

Interestingly, the lipid peroxidation in serum of obese rats perhaps may be related to the modulation of cholinergic receptors since nAChRs have shown an antioxidative effect and mAChR stimulation may generate reactive oxygen species [70]. However, the presented hypothesis requires further research and observations in both the frontal cortex and hippocampus.

## 5. Conclusions

Based on the functionally altered cholinergic markers in the frontal cortex and the hippocampus of DIO compared to CHOW rats, we suggest innovative perception into how obese condition can affect the cholinergic system and consequently the cognitive skills. Moreover, the modulation of certain cholinergic enzymes and/or receptors might be a useful therapeutic approach in the cerebral dysfunction related to obesity.

## Figures and Tables

**Figure 1 nutrients-14-01243-f001:**
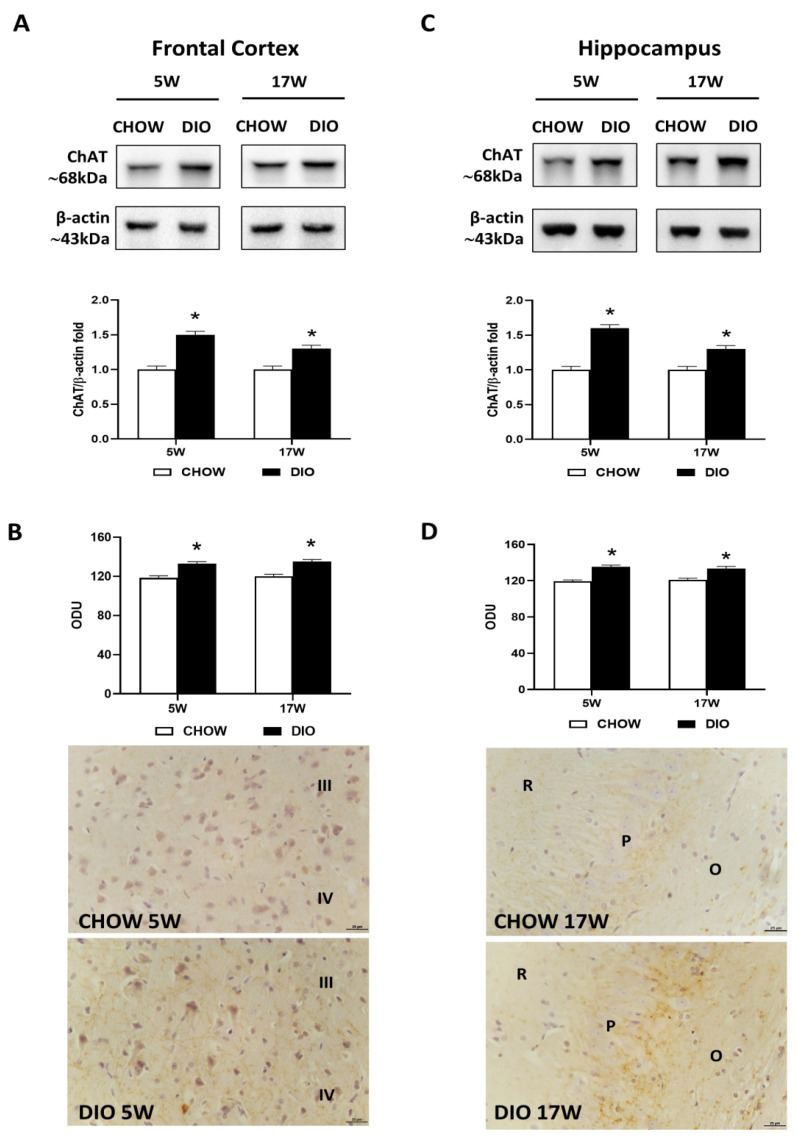
Western blot and immunohistochemistry of choline acetyltransferase (ChAT). Samples of the frontal cortex (**A**) and hippocampus (**C**) from rats fed with a standard diet (CHOW rats) and high-fat diet (DIO rats) for 5 and 17 weeks were immunoblotted with anti-ChAT. Graphs reported the densitometric data with CHOW as control, and β-actin was used as a reference loading protein. ChAT immunoreaction intensity in the frontal cortex (**B**) and the hippocampus (**D**) from CHOW and DIO after 5 and 17 weeks of a high-fat diet was measured in optical density units (ODU). Data are mean ± S.E.M. * *p* < 0.05 vs. age-matched CHOW rats. CHOW rats 5 weeks *n* = 8; DIO rats 5 weeks *n* = 8; CHOW rats 17 weeks *n* = 8; DIO rats 17 weeks *n* = 9. Representative pictures of CHOW and DIO rats frontal cortex after 5 weeks (**B**) and hippocampus, CA3 subfield, after 17 weeks (**D**) of diet. III, IV, third and fourth layers of frontal cortex respectively; O, *stratum oriens*; P, pyramidal neurons; R, *stratum radiatum*. Scale bar: 25 µm.

**Figure 2 nutrients-14-01243-f002:**
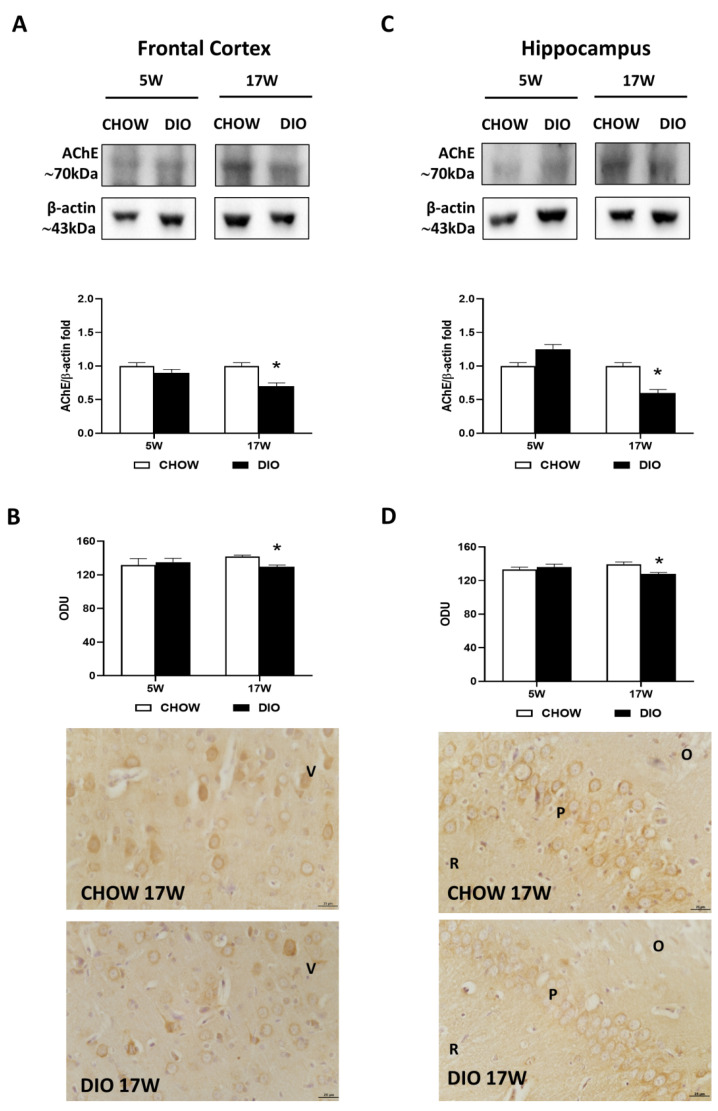
Western blot and immunohistochemistry of acetylcholinesterase (AChE). Samples of the frontal cortex (**A**) and hippocampus (**C**) from rats fed with a standard diet (CHOW rats) and high-fat diet (DIO rats) for 5 and 17 weeks were immunoblotted with anti-AChE. Graphs report the densitometric data with CHOW rats as control, and β-actin was used as a reference loading protein. AChE immunoreaction intensity in the frontal cortex (**B**) and the hippocampus (**D**) from CHOW and DIO rats after 5 and 17 weeks of a high-fat diet was measured in optical density units (ODUs). Data are mean ± S.E.M. * *p* < 0.05 vs. age-matched CHOW rats. CHOW rats 5 weeks *n* = 8; DIO rats 5 weeks *n* = 8; CHOW rats 17 weeks *n* = 8; DIO rats 17 weeks *n* = 9. Representative pictures of CHOW and DIO frontal cortex (**B**) and hippocampus, CA2 subfield, (**D**) after 17 weeks of diet. V, fifth layer of frontal cortex; O, *stratum oriens*; P, pyramidal neurons; R, *stratum radiatum*. Scale bar: 25 µm.

**Figure 3 nutrients-14-01243-f003:**
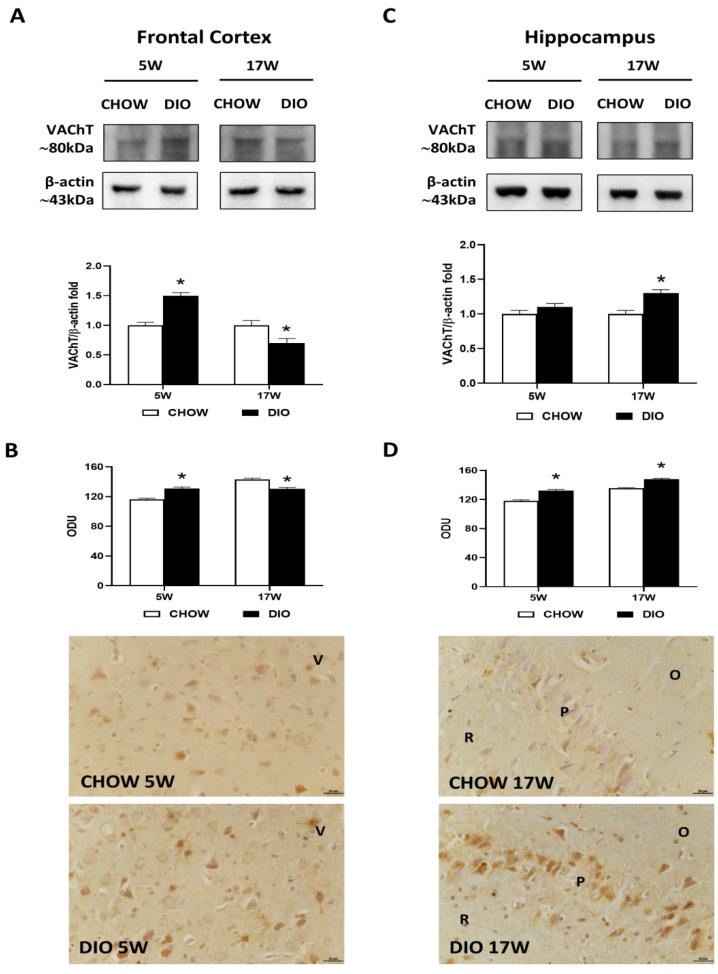
Western blot and immunohistochemistry of vesicular acetylcholine transporter (VAChT). Samples of the frontal cortex (**A**) and hippocampus (**C**) from rats fed with a standard diet (CHOW rats) and high-fat diet (DIO rats) for 5 and 17 weeks were immunoblotted with anti-VAChT. Graphs report the densitometric data with CHOW rats as control. ChAT membranes were stripped and incubated with anti-VAChT antibodies. β-actin control images were reused for illustrative purposes. VAChT immunoreaction intensity in the frontal cortex (**B**) and the hippocampus (**D**) from CHOW and DIO rats after 5 and 17 weeks of a diet was measured in optical density unit (ODU). Data are mean ± S.E.M. * *p* < 0.05 vs. age-matched CHOW rats. CHOW rats 5 weeks *n* = 8; DIO rats 5 weeks *n* = 8; CHOW rats 17 weeks *n* = 8; DIO rats 17 weeks *n* = 9. Representative pictures of CHOW and DIO rats frontal cortex after 5 weeks (**B**) and hippocampus, CA1 subfield, after 17 weeks (**D**) of diet. V, fifth layer of frontal cortex; O, *stratum oriens*; P, pyramidal neurons; R, *stratum radiatum*. Scale bar: 25 µm.

**Figure 4 nutrients-14-01243-f004:**
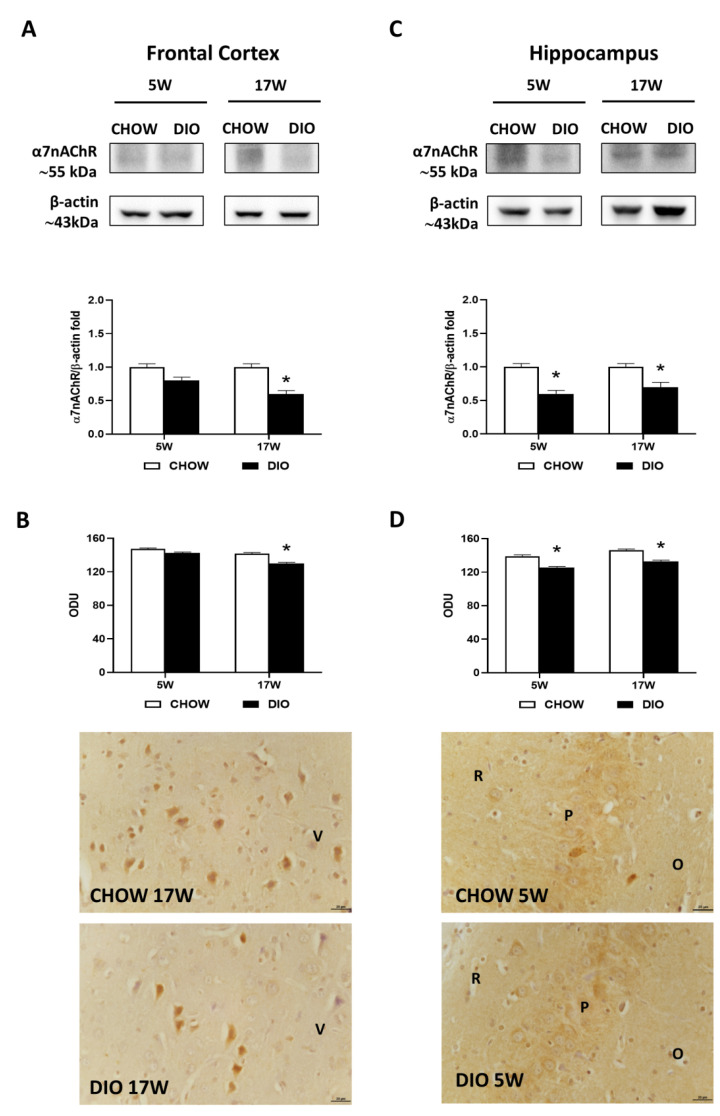
Western blot and immunohistochemistry of alpha7 nicotinic acetylcholine receptor (α7nAChR). Samples of the frontal cortex (**A**) and hippocampus (**C**) from rats fed with a standard diet (CHOW rats) and high-fat diet (DIO rats) for 5 and 17 weeks were immunoblotted with anti- α7nAChR. Graphs report the densitometric data with CHOW rats as control, and β-actin was used as reference loading protein. α7nAChR immunoreactions intensity in the frontal cortex (**B**) and the hippocampus (**D**) from CHOW and DIO rats after 5 and 17 weeks of a diet was measured in optical density unit (ODU). Data are mean ± S.E.M. * *p* < 0.05 vs. age-matched CHOW rats. CHOW rats 5 weeks *n* = 8; DIO rats 5 weeks *n* = 8; CHOW rats 17 weeks *n* = 8; DIO rats 17 weeks *n* = 9. Representative pictures of CHOW and DIO rats frontal cortex after 17 weeks (**B**) and hippocampus, CA3 subfield, after 5 weeks (**D**) of diet. V, fifth layer of frontal cortex; O, *stratum oriens*; P, pyramidal cells; R, *stratum radiatum*. Scale bar: 25 µm.

**Figure 5 nutrients-14-01243-f005:**
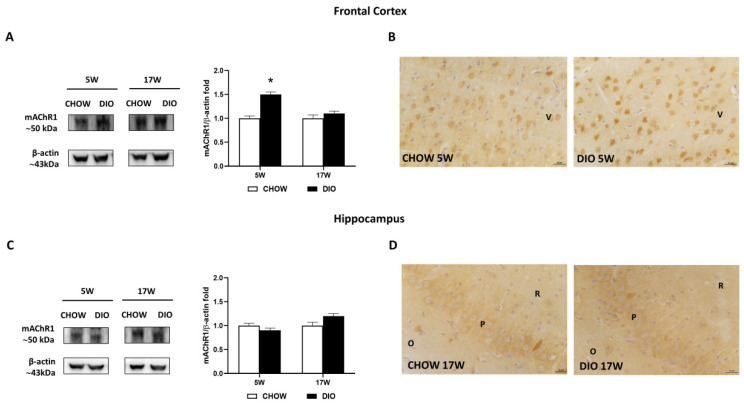
Western blot and immunohistochemistry of muscarinic acetylcholine receptor subtype 1 (mAChR1). Samples of frontal cortex (**A**) and hippocampus (**C**) from rats fed with standard diet (CHOW rats) and high-fat diet (DIO rats) for 5 and 17 weeks were immunoblotted with anti- mAChR1. Graphs report the densitometric data with CHOW rats as control, and β-actin was used as a reference loading protein. Data are mean ± S.E.M. * *p* < 0.05 vs. age-matched CHOW rats. CHOW rats 5 weeks *n* = 8; DIO rats 5 weeks *n* = 8; CHOW rats 17 weeks *n* = 8; DIO rats 17 weeks *n* = 9. Representative pictures of CHOW and DIO rats frontal cortex after 5 weeks (**B**) and hippocampus, CA3 subfield, after 17 weeks (**D**) of diet. V, fifth layer of frontal cortex; O, *stratum oriens*; P, pyramidal neurons; R, *stratum radiatum*. Scale bar: 25 µm.

**Figure 6 nutrients-14-01243-f006:**
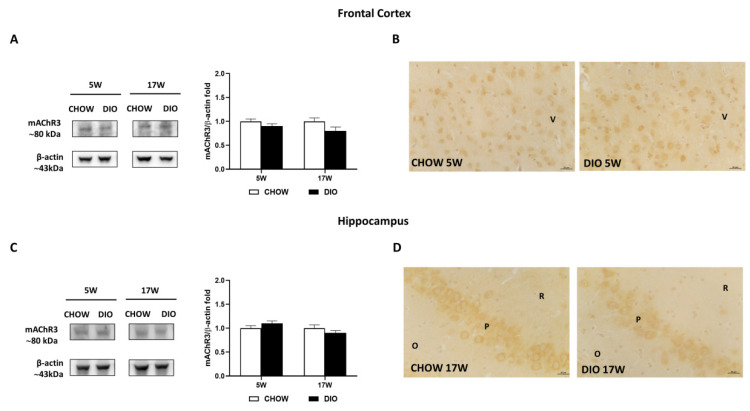
Western blot and immunohistochemistry of muscarinic acetylcholine receptor subtype 3 (mAChR3). Samples of frontal cortex (**A**) and hippocampus (**C**) from rats fed with standard diet (CHOW rats) and high-fat diet (DIO rats) for 5 and 17 weeks were immunoblotted using specific anti- mAChR3. Graphs report the densitometric data with CHOW rats as control, and β-actin was used as a reference loading protein. Data are mean ± S.E.M. CHOW rats 5 weeks *n* = 8; DIO rats 5 weeks *n* = 8; CHOW rats 17 weeks *n* = 8; DIO rats 17 weeks *n* = 9. Representative pictures of CHOW and DIO rats in frontal cortex after 5 weeks (**B**) and hippocampus, CA3 subfield, after 17 weeks (**D**) of diet. V, fifth layer of frontal cortex; O, *stratum oriens*; P, pyramidal neurons; R, *stratum radiatum*. Scale bar: 25 µm.

**Figure 7 nutrients-14-01243-f007:**
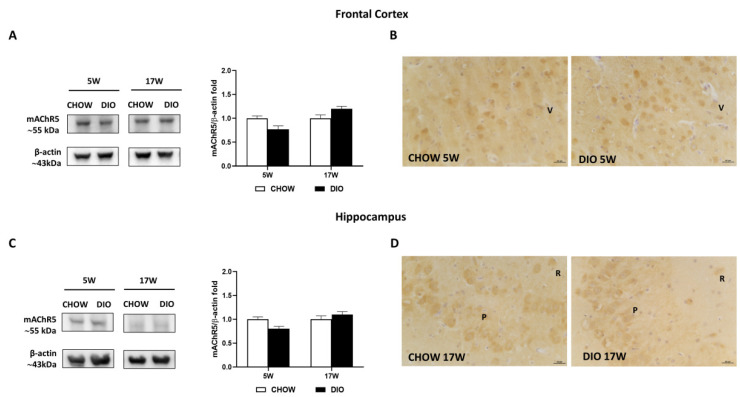
Western blot and immunohistochemistry of muscarinic acetylcholine receptor subtype 5 (mAChR5). Samples of the frontal cortex (**A**) and hippocampus (**C**) from rats fed with a standard diet (CHOW rats) and high-fat diet (DIO rats) for 5 and 17 weeks were immunoblotted using specific anti- mAChR5. Graphs report the densitometric data with CHOW rats as control, and β-actin was used as a reference loading protein. Data are mean ± S.E.M. CHOW rats 5 weeks *n* = 8; DIO rats 5 weeks *n* = 8; CHOW rats 17 weeks *n* = 8; DIO rats 17 weeks *n* = 9. Representative pictures of CHOW and DIO rats frontal cortex after 5 weeks (**B**) and hippocampus, CA3 subfield, after 17 weeks (**D**) of diet. V, fifth layer of frontal cortex; P, pyramidal neurons; R, *stratum radiatum*. Scale bar: 25 µm.

**Table 1 nutrients-14-01243-t001:** Primary antibodies used in immunohistochemistry (IHC) and Western blot (WB).

Antibody	Company and Cat. No	Dilution IHC	Dilution WB
Neuronal nuclei (NeuN)	Merck Millipore Cat. MAB377	1:500	1:1000
Neurofilament 200 kDa (NF)	Merck Millipore Cat. MAB526	1:500	1:1000
Choline acetyltransferase (ChAT)	Chemicon/Millipore Cat. AB144P	1:50	1:150
Acetylcholinesterase (AChE)	Santa Cruz Biotechnology Cat. sc6430	1:100	1:500
Vesicular acetylcholine transporter (VAChT)	Santa Cruz Biotechnology Cat. sc7717	1:100	1:500
Alpha7 nicotinic acetylcholine receptor (α7nAChR)	Santa Cruz Biotechnology Cat. sc5544	1:50	1:500
Muscarinic acetylcholine receptor subtype 1 (mAChR1)	Santa Cruz Biotechnology Cat. sc9106	1:50	1:500
Muscarinic acetylcholine receptor subtype 3 (mAChR3)	Santa Cruz Biotechnology Cat. sc7474	1:50	1:500
Muscarinic acetylcholine receptor subtype 5 (mAChR5)	Santa Cruz Biotechnology Cat. sc7479	1:50	1:500
Beta-actin (β-actin)	Sigma-Aldrich Cat. A2228	/	1:3000
Glyceraldehyde 3-phosphate dehydrogenase (GAPDH)	Sigma-Aldrich Cat. G9295	/	1:5000

## Data Availability

The data presented in this study are available on request from the corresponding author.

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
