# Peer review of "Obesity-Related Brain Cholinergic System Impairment in High-Fat-Diet-Fed Rats"

_nutrients, 2022, doi:10.3390/nu14061243_

Round 1
Reviewer 1 Report
Authors are instructed to include the Conceptual framework into the manuscript which is critical to this study.
Authors are also required to clarify, whether the model of HFD been validated already? If so they should provide the details of the validation.
Author Response
Author’s reply to the Reviewer 1
Authors are instructed to include the Conceptual framework into the manuscript which is critical to this study.
Thank you for your indications. In the introduction (lines 93-114), we better clarified the framework of the paper, focusing on the characterization of the cholinergic system alterations in the brain of obese male rats. In fact, despite the well-known global impact of overweight and of obesity in the brain dysfunction incidence, many aspects of this association, as the involvement of the cholinergic pathway, are still inconsistently defined. In our study, we aimed at establishing this connection by assessing the effects of obesity on brain cholinergic signaling in a rodent model of diet-induced obesity (DIO), which mimics common features of human obesity more accurately than other genetic models. The study was performed in cerebral areas such as the frontal cortex and the hippocampus, where cholinergic neurotransmission is widely represented. The expression of cholinergic markers, such as choline acetyltransferase (ChAT) and acetylcholinesterase (AChE), enzymes implicated in synthesizing and in breaking down of the acetylcholine (ACh) respectively, was examined. In addition, the levels of the vesicular acetylcholine transporter (VAChT), which mediates the packaging and transport of ACh, were examined. Different AChRs muscarinic acetylcholine receptors subtypes (mAChR1, mAChR3, and mAChR5) were explored because of their ubiquity and rich presence in the cerebral cortex and the hippocampus. Moreover, nicotinic alpha-7 receptor (α7nAChR) predominant and widely expressed at pre-and postsynaptic levels was analyzed.
Authors are also required to clarify, whether the model of HFD been validated already? If so they should provide the details of the validation.
Thank you for your question. The rodent model in which obesity is induced by a high-fat diet (Diet-Induced obesity; DIO) represents one of the most validated models to study obesity and its impact on end-organs damage. Several high energy diets have been utilized to precipitate obesity and related metabolic disorders in rodents, and we chose a standardized fat diet D12451, Research Diets (for reference: https://scholar.google.com/scholar?q=brunswick+%22d12451%22+%22research+diets%22&btnG=&hl=en)which is purchasable and then the results easily replicable. As reported by Archer and Mercer in 2007, “Rodent models of diet-induced obesity (DIO) mimic common human obesity more accurately than obese single-gene mutation lines, such as the ob/ob mouse. Nevertheless, the responses of these rats to solid and liquid obesogenic diets are very similar to those reported elsewhere, and this model of DIO has much to commend it as a vehicle for the mechanistic study of susceptibility to DIO, development, and reversal of obesity on solid and liquid diets and the response of peripheral and central energy balance systems to the development of obesity and the obesogenic diets themselves” [Archer ZA, Mercer JG. Brain responses to obesogenic diets and diet-induced obesity. Proc Nutr Soc. 2007 Feb;66(1):124-30. DOI: 10.1017/S0029665107005356.]. In 2010, Hariri and Thibault, in an important review demonstrated that “rats and mice show a similar relationship, they are considered an appropriate model for studying dietary obesity and clarify the consequences of changing the amount and type of dietary fats on weight gain, body composition, and adipose tissue cellularity and explores the contribution of genetics and sex as well as the biochemical basis, and the roles of hormones such as leptin, insulin, and ghrelin in animal models of dietary obesity” [Hariri N, Thibault L. High-fat diet-induced obesity in animal models. Nutr Res Rev. 2010 Dec; 23(2):270-99. DOI: 10.1017/S0954422410000168. Epub 2010 Oct 27. PMID: 20977819]. Data from PubMed indicate that in the last 10 years, more than 4000 papers used DIO rats as an animal model of obesity.
Our research group already used and validated DIO animal model as demonstrated by the following published works:
Cifani C, Micioni Di Bonaventura MV, Pucci M, Giusepponi ME, Romano A, DiFrancesco A, Maccarrone M, D'Addario C. Regulation of hypothalamic neuropeptides gene expression in diet induced obesity resistant rats: possible targets for obesity prediction? Front Neurosci. 2015 Jun 8;9:187;
Giusepponi ME, Kern M, Chakaroun R, Wohland T, Kovacs P, Dietrich A, Schön MR, Krohn K, Pucci M, Polidori C, Micioni Di Bonaventura MV, Stumvoll M, Blüher M, Cifani C, Klöting N. Gene expression profiling in adipose tissue of Sprague Dawley rats identifies olfactory receptor 984 as a potential obesity treatment target. Biochem Biophys Res Commun. 2018 Nov 2;505(3):801-806;
Micioni Di Bonaventura MV, Martinelli I, Moruzzi M, Micioni Di Bonaventura E,Giusepponi ME, Polidori C, Lupidi G, Tayebati SK, Amenta F, Cifani C, Tomassoni D. Brain alterations in high fat diet induced obesity: effects of tart cherry seeds and juice. Nutrients. 2020 Feb 27;12(3):623;
Cocci P, Moruzzi M, Martinelli I, Maggi F, Micioni Di Bonaventura MV, CifaniC, Mosconi G, Tayebati SK, Damiano S, Lupidi G, Amantini C, Tomassoni D, Palermo FA. Tart cherry (Prunus cerasus L.) dietary supplement modulates visceraladipose tissue CB1 mRNA levels along with other adipogenesis-related genes inrat models of diet-induced obesity. Eur J Nutr. 2021 Aug;60(5):2695-2707.
Martinelli I, Micioni Di Bonaventura MV, Moruzzi M, Amantini C, Maggi F,Gabrielli MG, Fruganti A, Marchegiani A, Dini F, Marini C, Polidori C, Lupidi G, Amenta F, Tayebati SK, Cifani C, Tomassoni D. Effects of Prunus cerasusL. Seeds and Juice on Liver Steatosis in an Animal Model of Diet-InducedObesity. Nutrients. 2020 May 4;12(5):1308.
Roy P, Martinelli I, Moruzzi M, Maggi F, Amantini C, Micioni Di BonaventuraMV, Cifani C, Amenta F, Tayebati SK, Tomassoni D. Ion channels alterations in the forebrain of high-fat diet fed rats. Eur J Histochem. 2021 Nov23;65(s1):3305.
Moruzzi M, Klöting N, Blüher M, Martinelli I, Tayebati SK, Gabrielli MG, RoyP, Micioni Di Bonaventura MV, Cifani C, Lupidi G, Amenta F, Tomassoni D. TartCherry Juice and Seeds Affect Pro-Inflammatory Markers in Visceral Adipose Tissue of High-Fat Diet Obese Rats. Molecules. 2021 Mar 5;26(5):1403.

Reviewer 2 Report
This study showed that the positive modulation of certain cholinergic enzymes and-or receptors may be a possible therapeutic strategy for the treatment of obesity-related cerebral dysfunction. In this study, the authors used widely used orthodox experimental methods. However, there are some concerns regarding some parts of the manuscript.
My specific major comment follows below:
1. There are some issues in experimental animal design that need to be explained clearly (2.1. Animal Handling).
(1) [Line113-115] The feed of the DIO group is marked with 35% carbohydrates, 45% lipid, and 20% protein. The feed of the DIO group was marked with 35% carbohydrate, 45% fat, and 20% protein. However, there is no detailed description of the feed composition of the CHOW group, and it is recommended to label it.
(2) [Line 112-128] How many rats were sacrificed at 5th and 17 the weeks respectively? In this study, did the number of experimental rats comply with the 3R principles (Refinement, Reduction, Replacement) in IACUC?
(3) It is unclear how many rats were allocated and finally analyzed in each group. Please, provide the number of subjects per group in all figures. It would be helpful to the readers.
2. Whether H&E staining was performed to observe tissue (frontal cortex and hippocampus) damage?
3. In Figure 2, the immunoblotting and immunoreaction of alpha7 nicotinic acetylcholine receptor 326 (α7nAChR). in (A) (C), the image is not clear, is there anything else image?
4. Please confirm whether the indication of the week in the IHC staining observation chart is correct (Figure 2).
5. Western blot results for the VAChT showed that the expression of VAChT was significantly higher in the frontal cortex of obese rats fed for 5 weeks (Figure 3A). On the contrary, it was found that VAChT was significantly lower after 17 weeks with HFD. Why? Please explain clearly in the discussion.
Author Response
Author’s reply to the Reviewer 2
Comments and Suggestions for Authors
This study showed that the positive modulation of certain cholinergic enzymes and-or receptors may be a possible therapeutic strategy for the treatment of obesity-related cerebral dysfunction. In this study, the authors used widely used orthodox experimental methods. However, there are some concerns regarding some parts of the manuscript.
My specific major comment follows below:
- There are some issues in experimental animal design that need to be explained clearly (2.1. Animal Handling).
Thanks for your observation. In paragraph 2.1, Animal handling, we answer at different issues.
- The feed of the DIO group is marked with 35% carbohydrates, 45% lipid, and 20% protein. The feed of the DIO group was marked with 35% carbohydrate, 45% fat, and 20% protein. However, there is no detailed description of the feed composition of the CHOW group, and it is recommended to label it.
Thank you for noticing this lack of details regarding 4RF18 Mucedola standard diet. We have now added that diet of CHOW rats consisted of the intake of 2.6 kcal/g, which was distributed among carbohydrates, proteins, and fats according to the following percentages: 72%, 21%, and 7% (Lines125-126).
(2) [Line 112-128] How many rats were sacrificed at 5th and 17 the weeks respectively? In this study, did the number of experimental rats comply with the 3R principles (Refinement, Reduction, Replacement) in IACUC?
Starting from a total of 36 male Wistar rats, after 5 weeks of a diet, were sacrificed n = 8 DIO rats and n=8 CHOW rats. The remaining DIO rats were fed with HFD for another 12 weeks for a total of 17 weeks of HFD, while the remaining CHOW rats were fed with the standard diet.In the groups of DIO rats, after 17 weeks of HFD, the rats that were resistant to increasing body weight (n= 3) were excluded from the experiment because the obese phenotype was not develop. Finally, after 17 weeks,n=8 of CHOW rats and n=9 of DIO rats were sacrificed.
We confirm that use of rats was based on the principle of the 3Rs, following the European regulation:the number and the experimental procedure involving animal care were under the principle of Refinement, Reduction, and Replacement.
(3) It is unclear how many rats were allocated and finally analyzed in each group. Please, provide the number of subjects per group in all figures. It would be helpful to the readers.
Thank you for your suggestion. In each figure legend, we have now clearly indicated the number of animals.
- Whether H&E staining was performed to observe tissue (frontal cortex and hippocampus) damage?
Thank you for your suggestion. Instead of H&E staining, longitudinal sections of the brain were processed for the Niss’l Staining, which allows for the evaluation of morphological features of the nervous system. Nissl stains use a variety of dyes (e.g., thionin, cresyl violet, fluorescent compounds) to show charged structures (Nissl bodies) in the soma of neurons and glia.The Nissl stain is most intense in nucleoli and the rough endoplasmic reticulum of neurons [From Netter's Atlas of Neuroscience (Third Edition), 2016.Techniques for studying the brain Charles Watson, George Paxinos, in The Brain, 2010].To show the results, we added a new supplementary figure 1, which showed the images of frontal cortex and hippocampus from CHOW and DIO rats at different ages.
- In Figure 2, the immunoblotting and immunoreaction of alpha7 nicotinic acetylcholine receptor 326 (α7nAChR). in (A) (C), the image is not clear, is there anything else image?
Thank you for your suggestion. We improved the resolution of the Western Blot pictures and we showed only the corresponding molecular weight of the α7nAChR band at 55kDa.
- Please confirm whether the indication of the week in the IHC staining observation chart is correct (Figure 2).
Thank you for your suggestion. We confirm the indication of the weeks for different IHC staining.
- Western blot results for the VAChT showed that the expression of VAChT was significantly higher in the frontal cortex of obese rats fed for 5 weeks (Figure 3A). On the contrary, it was found that VAChT was significantly lower after 17 weeks with HFD. Why? Please explain clearly in the discussion.
Thank you for the suggestion. In the discussion (Lines 397-404), we explained how the expression of VAChT was modulated and the related mechanisms.
The alterations of axonal fibers could explain the down-regulation of VAChT expression in the frontal cortex of 24-weeks old DIO rats, similarly to the VAChT modulation present both in the frontal cortex and in the hippocampus of older OZRs [Martinelli, I.; Tomassoni, D.; Roy, P.; Amenta, F.;Tayebati, S.K. Altered Brain Cholinergic and Synaptic Markers in Obese Zucker Rats. Cells 2021, 10, 2528. DOI: 10.3390/cells10102528]. Differently, an upregulation of VAChT, as observed in spontaneously hypertensive rats, represents an attempt to compensate the cholinergic impairment in the first stages of brain disfunction induced by hypertension [Tayebati, S.K.; Di Tullio, M.A.; Amenta, F. Vesicular acetylcholine transporter (VAChT) in the brain of spontaneously hypertensive rats (SHR): effect of treatment with an acetylcholinesterase inhibitor. Clin Exp Hypertens2008, 30, 732-43. DOI: 10.1080/10641960802580216; Tomassoni, D.; Catalani, A.; Cinque, C.; Di Tullio, M.A.; Tayebati, S.K.; Cadoni, A.; Nwankwo, I.E., Traini, E.; Amenta F. Effects of cholinergic enhancing drugs on cholinergic transporters in the brain and peripheral blood lymphocytes of spontaneously hypertensive rats. Curr Alzheimer Res2012, 9, 120-7. DOI: 10.2174/156720512799015118.].

Round 2
Reviewer 2 Report
In general, the experimental work presented is fine and the results are clearly presented.
Author's Response is corrected in the manuscript. I think this paper merits publication in Nutrients.